# Clinical Utility of Circulating Tumor Cells for Predicting Major Histopathological Response after Neoadjuvant Chemoradiotherapy in Patients with Esophageal Cancer

**DOI:** 10.3390/jpm12091440

**Published:** 2022-08-31

**Authors:** Xing Gao, Osbert Qi-Yao Leow, Chien-Hung Chiu, Ming-Mo Hou, Jason Chia-Hsun Hsieh, Yin-Kai Chao

**Affiliations:** 1Division of Thoracic Surgery, Chang Gung Memorial Hospital-Linkou, Chang Gung University, Taoyuan 333, Taiwan; 2Department of General Surgery, Erasmus Medical Center, 3015GD Rotterdam, The Netherlands; 3Division of Hematology and Oncology, Department of Internal Medicine, Chang Gung Memorial Hospital-Linkou, Chang Gung University, Taoyuan 333, Taiwan; 4Division of Hematology and Oncology, Department of Internal Medicine, New Taipei Municipal Tucheng Hospital, New Taipei City 236, Taiwan

**Keywords:** Circulating tumor cells, esophageal cancer, neoadjuvant chemoradiotherapy, major histopathological response, surgery as needed

## Abstract

Background: A “surgery as needed” approach may be offered to patients with esophageal cancer (EC) who achieve major histopathological response (MaHR) after neoadjuvant chemoradiotherapy (nCRT). However, the utility of clinical response assessment (CRE) for predicting histopathological response to nCRT remains limited. Circulating tumor cells (CTCs) hold promise as biomarkers of response to nCRT. Methods: We analyzed the clinical utility of post-nCRT CTCs, alone or in combination with CRE, in the prediction of MaHR. We defined MaHR as either the lack or a limited presence (≤10%) of vital residual tumor cells in the resected esophageal specimen in the absence of nodal involvement. Results: Of the 48 study patients, 27 (56%) achieved MaHR. Patients with MaHR had a significantly lower CTCs count compared with those without (3.61 ± 4.53 versus 6.83 ± 5.22 per mL of blood, respectively; *P* = 0.027). Using a cutoff for positivity of 5 CTCs per mL of blood, the combination of CTCs and CRE allowed achieving a negative predictive value for MaHR of 93% (95% confidence interval [CI] = 70–99%) along with a false negative rate of 5% (95% CI = 1–33%). Conclusion: CTCs count assessed in combination with CRE can potentially help identify patients with EC who achieved MaHR after nCRT.

## 1. Introduction

Neoadjuvant chemoradiotherapy (nCRT) followed by surgery is a standard first-line therapeutic strategy for patients with locally advanced esophageal cancer (EC) [1]. Recent years have witnessed an increasing incidence of no residual cancer (i.e., pathological complete response [pCR]) in esophagectomy specimens collected from patients who had undergone nCRT. This has called into question the necessity of esophagectomy [2,3,4,5].

Theoretically, the absence of residual cancer burden (0% vital residual tumor cells [VRTCs]) would not pose an indication for surgery. However, in a real-world scenario, the lack or a limited presence (≤10%) of VRTCs in the resected esophageal specimen in the absence of nodal involvement—a condition termed major histopathological response (MaHR)—is more commonly encountered. Patients who are capable of achieving MaHR following nCRT may benefit from a “surgery as needed” approach. Accordingly, tumors that will eventually develop in this patient subset are amenable to detection during active surveillance, when surgery still represents a curative option. Unfortunately, the identification of this patient group in the pre-surgical phase remains challenging. This can be attributed to the limited correlation between the pathological response and the standard clinical response evaluation (CRE)—consisting of endoscopic biopsies along with computed tomography (CT) and/or positron emission tomography (PET) imaging [6,7]. 

Over the last decade, the measurement of circulating tumor cells (CTCs) in the bloodstream has been repeatedly proposed as a biomarker of residual disease after curative treatment [8]. While data are promising in several solid cancers – including breast, pancreatic, colorectal, and esophageal malignancies [9,10,11,12], the questions as to whether the CTCs count may be clinically useful in the identification of EC patients with MaHR after nCRT remains unanswered. In the current study, we hypothesized that the measurement of CTCs in the post-nCRT phase would facilitate the selection of suitable candidates for an “active surveillance and surgery as needed” approach, i.e., the subset of patients with EC who had achieved MaHR. To this aim, CTCs were quantified following nCRT and their ability to predict MaHR was analyzed either alone or in combination with the results of CRE.

## 2. Materials and Methods

### 2.1. Study Design

All visits and procedures occurred at the Chang Gung Memorial Hospital (CGMH), Linkou (Taiwan). The study started in July of 2017 and concluded in July of 2019. Patients with locally advanced EC were eligible if they had undergone nCRT as first-line treatment followed by scheduled esophagectomy. CTCs were quantified once, immediately before post-nCRT surgery (within 12 weeks of nCRT completion). Patients were excluded if they had been diagnosed with other malignancies in the five years preceding the study initiation, the only exceptions being non-melanoma skin cancer and in situ cervical carcinoma. All participants provided written informed consent to the investigation, and approval was granted by the Institutional Review Board of the CGMH (reference number: 201600378B0). 

### 2.2. Neoadjuvant Chemoradiotherapy

The study patients received two different nCRT regimens, i.e., PF and CROSS. The PF regimen comprised 5-fluorouracil (5-FU; 1000 mg/m^2^ per day, administered by continuous infusion over 96 h from day 1 to day 4 and from day 29 to day 33) and cisplatin (75 mg/m^2^; given intravenously over 3 h on day 1 and day 29). An intended radiotherapy dose of 45 Gy was delivered from day 1 to day 35 (daily fraction = 1.8 Gy, five days per week). Treatment fields were designed to encompass the entire esophagus, regional lymphatics, and the lymphatic structures located in the celiac and pericardial regions, as well as in the supraclavicular fossa—provided that the dose delivered to normal tissues was tolerable. The CROSS regimen consisted of weekly administrations of carboplatin (with doses titrated to achieve an area under curve of 2 mg/mm/min) and paclitaxel (50 mg/m^2^ of body-surface area) given for five weeks in combination with concurrent radiotherapy (45 Gy in five fractions, five days per week).

### 2.3. Clinical Restaging after Neoadjuvant Chemoradiotherapy

Clinical restaging was planned at 5−7 weeks after completion of nCRT. Post-treatment evaluations comprised a thorough physical examination, endoscopic biopsies, thoracic and abdominal CT scans, and PET imaging. A complete response (CR) on endoscopic examination was defined by the absence of tumor cells in biopsy specimens. A minimum of three biopsy samples were obtained when suspicious lesions were present. When clearly identifiable lesions were lacking, random biopsies were taken from the preexisting tumor site. On examining CT images, a CR in the affected lymph nodes was considered to be present when shrinkage to less than 10 mm in the short axis was evident. With regards to PET scans, CR was defined as a decrease in fluorodeoxyglucose (FDG) accumulation in the primary tumor site and/or the affected lymph nodes to a level that was indistinguishable from the surrounding normal tissues. Diffuse FDG uptake within the radiation field in the absence of focal lesions was considered as radiotherapy-induced esophagitis and was a valid criterion for CR. When both endoscopic and radiological evaluations failed to identify residual disease based on the consensus of a multidisciplinary team, a clinical complete response (cCR) was considered to be present.

### 2.4. Surgical Resection 

Patients who had undergone clinical restaging were eligible for surgery if they had an American Society of Anesthesiologists (ASA) score ≤ 3, no signs of a tracheoesophageal fistula, and no evidence of metastatic spread to solid organs and/or distant nodes. The surgical approach consisted of a right side transthoracic esophagectomy with intrathoracic (Ivor Lewis procedure) or neck anastomosis (McKeown procedure). All of the study patients underwent two-field lymph node dissection. 

### 2.5. Pathological Assessment

Surgical specimens were opened longitudinally and fixed overnight in a 10% formalin solution. When residual tumors were present, representative sections were thoroughly examined for determining the maximum depth of invasion and the reciprocal relationships between the esophagus and the stomach. In the absence of gross tumors, ulcerated or fibrotic areas were sampled and representative sections were submitted to pathological examination. All slides were stained with hematoxylin and eosin. The amount of VRTCs was assessed in the total cancer area using a four-point scale, as follows: 0%, 1–10%, 11–50%, >50% [13].

### 2.6. Quantification of Circulating Tumor Cells by Flow Cytometry 

Figure 1a–f show a schematic representation of the procedure used for CTCs analysis, followed by Figure 1g–l demonstrating the gating strategy through scatter plots. Prior to quantification, CTCs were enriched in blood samples through a negative selection procedure aimed at depleting CD45^+^ leukocytes and inducing red blood cells (RBCs) lysis. In brief, blood specimens (6−8 mL) were collected by venipuncture and the first 2−4 mL were discarded to avoid contamination with epithelial cells and stored as backup samples. The remaining 4 mL were used to quantify CTCs. First, samples were exposed to RBC lysis buffer (155 mM NH4Cl, 14 mM NaHCO3, and 0.1 mM EDTA in a 10:1 ratio). Subsequently, CD45^+^ leukocytes were depleted using the EasySep™ Human CD45 Depletion Kit (catalogue number: 18259; STEMCELL Technologies Inc., Vancouver, Canada). Immunomagnetically enriched specimens spiked with OECM-1 cells—a human squamous cell carcinoma cell line (Food Industry Research and Development Institute, Hsinchu City, Taiwan)—underwent labeling with an Alexa Fluor 488-conjugated anti-EpCAM antibody (catalogue number: 5198S; Cell Signaling Technology, Danvers, MA, USA). The Hoechst 33342 dye (catalogue number: 62249; Thermo Fisher Scientific, Waltham, MA, USA) was used for nuclear staining. 

The gating strategy contains six steps. (Figure 1g–l) First, the Hoechst^+^ cells are gated in 2 mL samples from all events to avoid cell debris and fragmentations after the negative selection process. (Figure 1g) Then, singlet cells were gated to avoid false positive results due to cell aggregation. (Figure 1h,i) CD45^+^ cells were then excluded again to avoid residual white blood cell contamination (Figure 1j). Before CTC enumeration, we independently gated EpCAM^+^ (and its isotype^+^) cells. (Figure 1k,l) Finally, CTCs—which were negative for CD45 but stained positive for both EpCAM and Hoechst 33342—were quantified on a CytoFLEX flow cytometer (Beckman Coulter, San Diego, CA, USA). The CTC count is defined as the number of EpCAM^+^ cells (Figure 1k) minus the number of cells gated using its isotype (Figure 1l).

### 2.7. Statistics 

Continuous variables are expressed as means ± standard deviations (SD) and differences were compared using Student’s *t*-tests or Kruskal-Wallis tests, as appropriate. Categorical variables were analyzed between groups using chi-square tests. Fisher’s exact tests were applied when the expected cell count was lower than 5. The associations between CTCs count (measured per mL of blood) and pCR/MaHR were modeled by univariate logistic regression analysis. Four different cutoff values for CTCs were tested (≥0, ≥3, ≥5, and ≥7 per mL of blood). The results were expressed as odds ratios (ORs) and 95% confidence intervals (CIs). The false negative rate (FNR), specificity, and negative predictive value (NPV) were calculated for MaHR versus no MaHR. The Wilson’s procedure was used to calculate the 95% CIs. No correction for continuity was applied [14]. Overall survival (OS) was defined as the time elapsed from the date of nCRT initiation to the date of death. Censoring was performed on the date of the last follow-up (i.e., administrative censoring). Survival probabilities were graphically represented with Kaplan-Meier curves (log-rank test). Data management and analyses were carried out using SPSS for Windows (IBM, Armonk, NY, USA). All hypothesis testing was two-tailed, with statistical significance defined as a *P* value < 0.05.

## 3. Results

### 3.1. Patient Characteristics

The general characteristics of the 48 study patients (43 men and 5 women; mean age: 56.10 ± 12.5 years) are summarized in Table 1. The vast majority of the study participants (46/48, 95.83%) were diagnosed with esophageal squamous cell carcinoma (ESCC), and most had clinical stage III−IVa disease. The mean pretreatment tumor length was 5.54 ± 3.9 cm. With regards to nCRT, the CROSS regimen was the most commonly used chemotherapy combination. On analyzing the response to nCRT, 21 patients had evidence of cCR. The results of pathology revealed that 27 (56.25%) and 21 (43.75%) patients achieved MaHR and pCR, respectively. Two patients had irresectable tumors upon surgery and were scored ypT4Nx, thus no MaHR.

### 3.2. Quantification of Circulating Tumor Cells and Determination of the Optimal Cutoff Value 

Thirty-seven (77%) patients had CTCs in mL of blood, with a mean CTCs count of 5.02 per mL of blood. Patients with evidence of MaHR had a significantly lower CTCs count compared with those without (3.61 ± 4.53 versus 6.83 ± 5.22 per mL of blood, respectively; *P* = 0.027). While a similar trend was observed for patients with and without pCR (3.83 ± 4.98 versus 5.94 ± 5.01 per mL of blood, respectively, *P* = 0.153), the difference failed to reach the threshold for statistical significance. On analyzing four different cutoff values for CTCs (≥0, ≥3, ≥5, and ≥7 per mL of blood), two thresholds (≥5, and ≥7 per mL of blood) were significantly associated with the achievement of MaHR. Further analyses revealed that a cutoff of ≥5 CTCs per mL of blood was associated with the lowest *P* value (Table 2); this point was therefore selected for further analyses. 

### 3.3. Survival Analysis

The median OS in the entire study cohort was 33.56 months. The 3-year OS rate in patients with ≥5 CTCs per mL of blood was significantly lower than that observed in those with less than 5 CTCs per mL of blood (24% versus 68%, respectively; *P* = 0.0019; Figure 2a). The same results were found in patients with no pCR or no MaHR, versus patients without a pCR or MaHR (Figure 2b,c). The adverse prognostic significance of the presence of ≥5 CTCs per mL of blood was also evident when the achievement of either MaHR or pCR was taken into account. Specifically, the 3-year OS of patients who achieved MaHR and had less than 5 CTCs per mL of blood was 90% versus 57% of those with evidence of MaHR and ≥5 CTCs per mL of blood (*P* = 0.043; Figure 2d). In addition, the 3-year OS of patients who achieved pCR and had less than 5 CTCs per mL of blood was 93% versus 50% of those with evidence of pCR and ≥5 CTCs per mL of blood (*P* = 0.017; Figure 2e).

### 3.4. Accuracy of Clinical Response Assessment and Circulating Tumor Cells in the Detection of MaHR 

Table 3 illustrates the accuracy for the detection of MaHR using CRE tools, the CTCs count, and their combination. The NPVs for MaHR associated with the use of CRE tools and the presence of less than 5 CTCs per mL of blood were 81% (95% CI = 60−92%) and 71% (95% CI = 53−85%), respectively. When these two parameters were analyzed simultaneously, the NPV increased to 93% (95% CI = 70−99%, Table 3). CRE tools yielded false-negative results in 4 of the 21 patients (FNR = 29%; 95% CI = 8−40%) who had evidence of major residual disease (i.e., no-MaHR). However, the FNR decreased to 5% (95% CI = 1−33%) when the CTC count was incorporated into the prediction model. 

## 4. Discussion

This study suggests that CTCs quantification—together with traditional CRE—can contribute to the prediction of nCRT response in patients with EC. In particular, CTCs quantified in the post-nCRT phase may be clinically useful for identifying patients with EC who achieved a MaHR and are likely to benefit from an “active surveillance and surgery as needed” approach. Patients who achieved MaHR after nCRT had a significantly lower number of CTCs in the bloodstream. Interestingly, when CTCs count was analyzed in combination with the results of standard CRE, we obtained a NPV for MaHR as high as 93%. We also observed that the presence of ≥5 CTCs per mL of blood was associated with less favorable OS figures even when either MaHR or pCR were obtained. Taken together, these findings demonstrate the clinical value of CTCs as a biomarker for predicting not only the achievement of MaHR after nCRT but also the survival outcomes of patients with EC. 

There is increasing evidence that standard CRE does not possess sufficient accuracy for predicting the presence of residual EC after nCRT [6,7]. Additionally, the agreement between clinical and pathological results remains too low to guide the delicate decision of delaying or avoiding surgery [15,16]. While predicting the response to nCRT is still an unmet clinical need, biomarkers have great potential to improve the existing accuracy. While we found that CTCs alone did not outperform CRE in terms of NPV (71% versus 81%, respectively), the NPV increased to 93% when the two assessments were analyzed in combination. Our current findings confirm and expand previous pilot observations on the predictive value of CTCs in other solid tumors (e.g., locally advanced rectal cancer) [17,18,19]. However, only two previous studies specifically focused on biomarker-based prediction of response to neoadjuvant therapy in EC [20,21]. On analyzing mesenchymal CTCs, Chen et al. [21] have previously detected their presence in 70.6% of patients with progressive and stable disease versus 36.4% of those who achieve a complete or partial response. Our findings are consistent with a link between a reduced CTCs count and more favorable responses to nCRT. 

As for the predictive accuracy, this study is the first to present results that are compatible with those of the pre-SANO trial [22], where only 10% of major residual tumors had false-negative results on repeated endoscopic bite-on-bite biopsy and fine-needle aspiration of suspicious lymph nodes. However, while endoscopic assessments are technically demanding and limited by their invasive nature, CTCs quantification is attractive because the testing is cost-efficient and non-invasive. If independently validated in future studies, this peripheral biomarker will likely have a major impact in the multidisciplinary management of patients with EC. 

In our cohort, we found that 13 patients who achieved MaHR didn’t have a cCR or had ≥5 CTCs per mL of blood, which resulted in a low specificity of 52% despite the high NPV and low FNR. This could be due to an inaccurate classification of cCR because PET/CT imaging might not be always able to distinguish between substantial residual disease and nCRT-induced inflammatory changes [23]. As for the high CTC count, a spurious increase in CTCs can be attributed to circulating epithelial cells that are falsely detected as tumor cells by CTCs assays [24]. Another issue is that some patients who had evidence of MaHR on pathology may have developed systemic metastases that were outside the resected specimen. This possibility is exemplified by Figure 2d, where patients (*n* = 7) with ≥5 CTCs per mL of blood had unfavorable OS figures despite having achieved MaHR.

In the literature, "typical CTCs” are generally defined as cells expressing the surface antigen EpCAM (EpCAM^+^ CTCs). Nevertheless, CTCs might undergo an epithelial-to-mesenchymal transition in metastatic disease after which their EpCAM expression is downregulated. These EpCAM^−^ CTCs are considered as “atypical CTCs” and can also be analyzed using our protocol [25,26]. However, given the uncertain clinical significance of EpCAM^−^ CTCs in patients with esophageal cancer, we only defined CTCs as Hoechst^+^CD45^−^EpCAM^+^ cells in this study. 

Several limitations to this study must be noted. First, a small sample size and a lack of external validation are significant limitations, for which the power of data was compromised to some extent. Our findings may therefore only be considered as a proof of concept, which needs further validation in larger cohorts before changes in clinical practice can be suggested. Second, the vast majority of the study participants had a diagnosis of ESCC. Therefore, our results might not be generalizable to other histological types. Finally, we only measured CTCs post-nCRT and thus have no information on the dynamic changes in CTC numbers. For future analyses, we hope to incorporate CTCs quantification at each treatment stage in order present a more accurate prediction model.

## 5. Conclusions

The combination of CTCs quantification and CRE can potentially help identify patients with EC who have achieved MaHR, i.e., the subset more likely to benefit from a “surgery as needed” approach following nCRT. These findings need further validations before changes in clinical practice can be suggested.

## Figures and Tables

**Figure 1 jpm-12-01440-f001:**
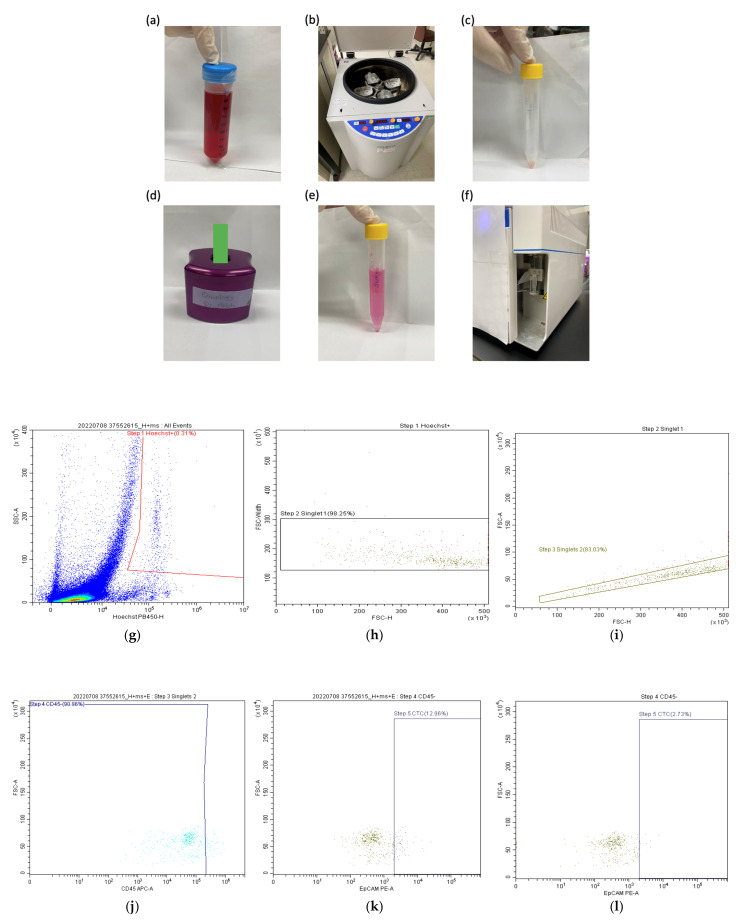
(**a**–**f**) Schematic representation of the procedure used to quantify circulating tumor cells. (**a**) Whole blood (4 mL) was mixed with red cell lysis buffer (40 mL) in a 50 mL tube, followed by incubation for 5 min. (**b**) The mixture was centrifuged at 400 g for 10 min and the supernatant was removed. (**c**) The settled cell pellet was aspirated, moved to a new 15 mL tube, and washed with RPMI medium (2 mL). After centrifugation at 400 g for 10 min, the supernatant was removed. Buffer N (10 μL) followed by incubation for 5 min and N beads (20 μL) followed by incubation for 5 min were consecutively added. (**d**) RPMI medium (2 mL) was added to resuspended cells, and the mixture was placed in a magnetic field to remove leukocytes. (**e**) The tube was washed with RPMI medium (6 mL) and the cell filtrate was transferred a new 15 mL tube. After centrifugation at 400 g for 10 min, the supernatant was removed. (**f**) The final step consisted of flow cytometric staining by EpCAM and its isotype, each derived from 2 mL whole blood. (**g**–**l**): The gating strategy contains six steps: (**g**) the Hoechst^+^ cells are gated from all events to avoid cell debris and fragmentations. (**h**–**i**) We then gated singlet cells to avoid false positive results due to cell aggregation. (**j**) CD45^+^ cells were then excluded again to avoid residual white blood cell contamination. Before CTC enumeration, we independently gated (**k**) EpCAM^+^ cells and (**l**) the corresponding isotype.

**Figure 2 jpm-12-01440-f002:**
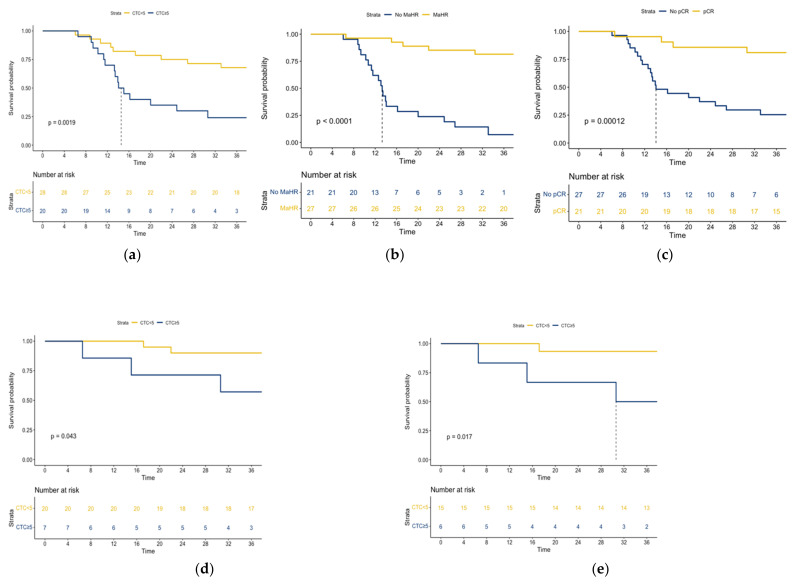
Kaplan-Meier plots of overall survival in patients with (**a**) ≥5 CTCs per mL of blood versus less than 5 CTCs per mL of blood, (**b**) MaHR versus no MaHR and (**c**) pCR versus no pCR in the entire study cohort (*n* = 48). Kaplan-Meier plots of overall survival in patients with ≥5 CTCs per mL of blood versus less than 5 CTCs per mL of blood in (**d**) patients who achieved MaHR (*n* = 27) and (**e**) patients who achieved pCR (*n* = 21).

**Table 1 jpm-12-01440-t001:** General characteristics of the study patients. Abbreviations: pCR, pathological complete response; MaHR, major histopathological response; PF, 5-fluorouracil/cisplatin; CROSS, carboplatin/paclitaxel; nCRT, neoadjuvant chemoradiotherapy; cCR, clinical complete response; CTCs, circulating tumor cells (post nCRT). Data are presented as means ± standard deviations or counts and percentages, as appropriate.

	Entire Cohort *n* = 48	pCR *n* = 21	No pCR *n* = 27	*P*	MaHR *n* = 27	No MaHR *n* = 21	*P*
**Sex**				*1.0*			*0.858*
**Male**	**43 (90)**	19 (90)	24 (89)		24 (89)	19 (90)	
**Female**	**5 (10)**	2 (10)	3 (11)		3 (11)	2 (10)	
**Age, years**	**56 ± 12.5**	57.76 ± 10.11	54.81 ± 8.49	*0.278*	57.56 ± 10.41	54.24 ± 7.33	*0.221*
**Clinical stage**				*0.383*			*0.683*
**II**	**6 (13)**	4 (19)	2 (7)		4 (15)	2 (10)	*0.221*
**III−IVa**	**42 (87)**	17 (81)	25 (93)		23 (85)	19 (90)	*0.683*
**Tumor length, cm**	**5.54 ± 3.9**	5.02 ± 3.42	6.19 ± 2.24	*0.171*	4.95 ± 3.14	6.57 ± 2.22	*0.058*
**Chemotherapy regimen**				*0.897*			*0.498*
**PF**	**11 (23)**	5 (24)	6 (22)		5 (19)	6 (26)	
**CROSS**	**37 (77)**	16 (76)	21 (78)		22 (81)	15 (74)	
**cCR**	**21(44)**	14 (67)	7 (26)	*0.005*	17 (63)	4 (19)	*0.002*
**CTCs count per mL of blood**	**5.02 ± 1.42**	3.83 ± 4.98	5.94 ± 5.01	*0.153*	3.61 ± 4.53	6.83 ± 5.22	*0.027*
**Pathological T stage**				*<0.001*			*<0.001*
**ypT0**	**24 (50)**	21 (100)	3 (11)		21 (78)	3 (14)	
**ypT1**	**5 (11)**	0 (0)	5 (19)		3 (11)	2 (10)	
**ypT2**	**3 (6)**	0 (0)	3 (11)		2 (7)	1 (4)	
**ypT3**	**14 (29)**	0 (0)	14 (52)		1 (4)	13 (62)	
**ypT4**	**2 (4)**	0 (0)	2 (7)		0 (0)	2 (10)	
**Pathological N stage**				*0.007*			*<0.001*
**ypN0**	**38 (79)**	21 (100)	17 (63)		27 (100)	11 (52)	
**ypN+**	**8 (17)**	0 (0)	8 (30)		0 (0)	8 (38)	
**ypNx**	**2 (4)**	0 (0)	2 (7)		0 (0)	2 (10)	

**Table 2 jpm-12-01440-t002:** Determination of the optimal cutoff value for circulating tumor cells in the assessment of response to neoadjuvant chemoradiotherapy.

		pCR			MaHR		
CTCs Count per mL of Blood	n (%)	OR	95% CI	*P*	OR	95% CI	*P*
≥ 0 = 0	37 (77) 11 (23)	1 2.88	0.71–11.62	*0.138*	1 2.526	0.58–11.05	*0.218*
≥ 3 < 3	29 (60) 19 (40)	1 2.16	0.66–7.10	*0.205*	1 2.971	0.85–10.44	*0.089*
≥ 5 < 5	20 (42) 28 (58)	1 2.69	0.80–9.04	*0.109*	1 4.643	1.36–15.91	*0.023*
≥ 7 < 7	11 (23) 37 (77)	1 2.53	0.58–11.05	*0.218*	1 4.923	1.11–21.82	*0.036*

Abbreviations: pCR, pathological complete response; MaHR, major histopathological response; CTCs, circulating tumor cells; OR, odds ratio; CI, confidence interval.

**Table 3 jpm-12-01440-t003:** Accuracy of circulating tumor cells and clinical response evaluation in the prediction of major histopathological response. Abbreviations: MaHR, major histopathological response; cCR, clinical complete response; FNR, false negative rate; NPV, negative predictive value; CTCs, circulating tumor cells. Data are presented as counts and percentages.

	No MaHR	MaHR	FNR	Specificity	NPV	Accuracy
No cCR	17	10	4/21 (29%)	17/27 (63%)	17/21 (81%)	34/48 (71%)
cCR	4	17				
CTC count ≥5 per mL of blood	13	7	8/21 (38%)	20/27 (74%)	20/28 (71%)	33/48 (69%)
CTC count <5 per mL of blood	8	20				
Non-cCR or CTC count ≥5 per mL of blood	20	13	1/21 (5%)	14/27 (52%)	14/15 (93%)	34/48 (71%)
cCR and CTC count <5 per mL of blood	1	14				

## Data Availability

The data presented in this study are available on request from the corresponding author.

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
