# Peer review of "Clinical Utility of Circulating Tumor Cells for Predicting Major Histopathological Response after Neoadjuvant Chemoradiotherapy in Patients with Esophageal Cancer"

_jpm, 2022, doi:10.3390/jpm12091440_

Round 1

Reviewer 1 Report

The authors have addressed an important unmet need in the era of multimodal therapy in esophageal cancer. Till now, we are still lacking a predictive biomarker for patient selection in case of complete response after neoadjuvant therapy in esophageal cancer.

In this study, the authors used flow cytometry for CTC detection in patients with esophageal cancer. Their results demonstrate that CTC is a predictive biomarker for the achievement of MaHR and favorable overall survival.  Furthermore, They incorporated liquid biopsy (CTC) with histopathological evaluation as a tool to stratify patients who might benefit from a watch and wait for strategy after MaHR representing an important step towards personalized therapy. The manuscript is well-written and informative.

I have the following questions and concerns:

  1. The author reports that CTC quantification was applied before and after neoadjuvant therapy. However, in the result and discussion sections, it is not clear whether pre-treatment or after-treatment CTCs were associated with MaHR and OS.
  2. Did the CTC counts increase or decrease during multimodal therapy and did the increase or decrease predict MaHR, pCR, or Survival?
  3. Can you please comment on why CTC counts were associated with MaHR status but not pCR? Even when taking the cut off ( ≥ 7 cells/ 7.5ml )? According to Table 1. 21 patients were in common between the pCR and  MaHR groups (21 patients with PCR plus 3 x ypT1, 2 x ypT2 and 1x ypT3).
  4. There are many methods for CTC detection in esophageal cancer. Please comment on the validation of flow cytometry compared to other well-established techniques like Cellsearch.   For example,  DOI: 10.1097/SLA.0000000000001130, demonstrated a CTC detection rate of 20% in esophageal adenocarcinoma and 10% in squamous cell carcinoma. Which are much lower than those described in this study.
  5. In Table 1: Patients with UICC III and IVa were incorporated as one group. Was there a difference in CTC count between IVa and stages II or III? Since stage IVa represents most likely the patients with unresectable disease at the time of exploration.
  6. A small correction in Table 2. (≥0, =0) should be corrected to (≥1, =0).

Author Response

The authors have addressed an important unmet need in the era of multimodal therapy in esophageal cancer. Till now, we are still lacking a predictive biomarker for patient selection in case of complete response after neoadjuvant therapy in esophageal cancer.

In this study, the authors used flow cytometry for CTC detection in patients with esophageal cancer. Their results demonstrate that CTC is a predictive biomarker for the achievement of MaHR and favorable overall survival.  Furthermore, They incorporated liquid biopsy (CTC) with histopathological evaluation as a tool to stratify patients who might benefit from a watch and wait for strategy after MaHR representing an important step towards personalized therapy. The manuscript is well-written and informative.

I have the following questions and concerns:

  1. The author reports that CTC quantification was applied before and after neoadjuvant therapy. However, in the result and discussion sections, it is not clear whether pre-treatment or after-treatment CTCs were associated with MaHR and OS.

Thank you for your comments. CTCs were only quantified once post-nCRT, right before surgery. In the methods section, we stated that “CTCs were quantified immediately before surgery and within 12 weeks of nCRT completion”. Sorry for the misleading sentence. We changed this sentence to prevent further misunderstandings: CTCs were quantified once, immediately before surgery (within 12 weeks of nCRT completion). We have added this explanation to Table 1 as well.

  1. Did the CTC counts increase or decrease during multimodal therapy and did the increase or decrease predict MaHR, pCR, or Survival?

We apologize for the misunderstanding, as stated above. Unfortunately, CTCs at diagnosis are not available for this study group and we only have one post-nCRT CTC measure. The prediction of pathological response is based on the quantification of this single measure. We have added this to the limitations of our sutdy.

  1. Can you please comment on why CTC counts were associated with MaHR status but not pCR? Even when taking the cut off ( ≥ 7 cells/ 7.5ml )? According to Table 1. 21 patients were in common between the pCR and MaHR groups (21 patients with PCR plus 3 x ypT1, 2 x ypT2 and 1x ypT3).

Thank you for this insightful comment. First of all, we apologize for a typo throughout our manuscript. The correct unit of CTC quantification should be cells/mL instead of cells/7.5 mL. We have corrected this in our manuscript.

Our current CTC quantification/searching method is not advanced enough yet to detect cancer free patients (pCR), not even when considering a higher cut off value of 7 cells/mL. However, we can predict patients who have no cancer or a small amount of cancer cells (MaHR). For esophageal cancer, the prediction of pCR after nCRT is an unfulfilled demand but serological biomarkers are starting to show promising results. In the future, we hope to incorporate more case numbers and dynamic CTC changes, as you suggested above, in order to present a more accurate selection.

  1. There are many methods for CTC detection in esophageal cancer. Please comment on the validation of flow cytometry compared to other well-established techniques like Cellsearch.   For example, DOI: 1097/SLA.0000000000001130, demonstrated a CTC detection rate of 20% in esophageal adenocarcinoma and 10% in squamous cell carcinoma. Which are much lower than those described in this study.

Thank you for this comment. Although Cellsearch is the only FDA approved technique, but its detection rate of metastatic cancer does not exceed 50%.1,2

Quoted from Allard et al: “In 1316 blood samples from patients with metastatic breast cancer, CTTs can only be detected in 489 samples (26%). Cellsearch, although it is the only FDA-approved platform for CTC detection, the detection rate is unsatisfactory given the advances of CTC detection technology.”1 Many techniques have already successfully challenged the detection rate of Cellsearch since its launch in 2004.3 It has been demonstrated that a negative selection-based strategy (used in our study) exhibits a better efficiency in the detection of CTCs compared with positive enrichment techniques (Cellsearch). The first one uses an epithelial marker-independent technique to target and remove other cell types and leaving CTCs untouched, while the latter one only uses specific anti-epithelial cell adhesion molecules to target CTCs, a method prone to missing CTCs.4

1Allard, W. Jeffrey, et al. "Tumor cells circulate in the peripheral blood of all major carcinomas but not in healthy subjects or patients with nonmalignant diseases." Clinical cancer research 10.20 (2004): 6897-6904

2Andree, Kiki C., Guus van Dalum, and Leon WMM Terstappen. "Challenges in circulating tumor cell detection by the CellSearch system." Molecular oncology 10.3 (2016): 395-407.”

3Hsieh, Jason Chia-Hsun, and Tyler Ming‐Hsien Wu. “The selection strategy for circulating tumor cells (CTCs) isolation and enumeration: Technical features, methods, and clinical applications.” London: IntechOpen, 2016

4Xu, Yan, et al. "Circulating tumor cell detection: A direct comparison between negative and unbiased enrichment in lung cancer." Oncology letters 13.6 (2017): 4882-4886.

  1. In Table 1: Patients with UICC III and IVa were incorporated as one group. Was there a difference in CTC count between IVa and stages II or III? Since stage IVa represents most likely the patients with unresectable disease at the time of exploration.

There were only 2 patients with cstage IVa in our cohort. One had a pCR with negative CTCs and the other had substantial residual disease (13 CTCs/mL blood). Since stage IVa is commonly given neoadjuvant treatment in Asia, we did not divide them at baseline.

  1. A small correction in Table 2. (≥0, =0) should be corrected to (≥1, =0).

CTCs are quantified in a 2mL serum (Figure 1). When there is only 1 CTC in the 2mL serum, the unit would be 0.5cells /mL. Which is why we used (≥0, =0) in Table 2.

Reviewer 2 Report

I think this is a very valuable paper.

I think it deserves to be published, but I would like to know the following information.

Is the value of CTCs the same across carcinomas? In this study, the mean is 5.02 per 7.5 ml.

Is this value only for this experimental system? Thank you in advance.

Furthermore, is the cutoff value for CTCs determined in this study the same as for other carcinomas, such as breast cancer, pancreatic cancer, and colorectal cancer?

I believe that the CTCs value is important, but I suspect that the rate of change in the CTCs value before and after nCRT may be more important as a predictor of the effect of nCRT. Thank you in advance.

Table 1: What does y stand for in ypT and ypN? Please let us know.

Translated with www.DeepL.com/Translator (free version)

Author Response

AUTHORS’ RESPONSES TO COMMENTS FROM REVIEWER #2

I think this is a very valuable paper.

I think it deserves to be published, but I would like to know the following information.

  1. Is the value of CTCs the same across carcinomas? In this study, the mean is 5.02 per 7.5 ml.

Thank you for the appreciation of our paper. First of all, we apologize for a typo throughout our manuscript. The correct unit of CTC quantification should be cells/mL instead of cells/7.5 mL. We have corrected this in our manuscript.

To answer your question, there were only 2 patients with adenocarcinoma in our cohort. One had a pCR with 4.5 CTCs/mL and the other had substantial residual disease (15 CTCs/mL). The sample size of adenocarcinoma at our institution in Taiwan is too small to do separate analyses.

  1. Is this value only for this experimental system? Thank you in advance. Furthermore, is the cutoff value for CTCs determined in this study the same as for other carcinomas, such as breast cancer, pancreatic cancer, and colorectal cancer?

Thank you for the insightful comments. CTC counts and cut off values vary significantly depending on the protocol, device, or strategy of CTC isolation. Therefore, the values presented in this study are not representable for other systems. However, our method uses tools available in most laboratories, so it can be easily replicated in others labs as well.

Theoretically, different tumor characteristics and especially different carcinomas should have different cutoff values, resulting in different sensitivity and specificity. The Cellsearch system, for example, has different cutoffs.1 However, extremely large sample sizes are needed to accurately detect those values. We have detected smaller cut off values in breast and lung cancer using the same quantification method (3 cells/mL), but the case numbers were too small to standardize these cut offs. 2,3

In the present study, we have tested multiple cutoff values of CTC counts, and two cutoffs (<5 and <7 cells/mL blood) were found to be significant in the association with MaHR. The same value (<5 cells/ml blood) has been previously described using comparable methods 4,5,6 Because of the limited sample size and no standard cutoff value is available for esophageal cancer, we chose <5 cells/ml blood based on its statistical significance and reported clinical significance in other literature.

1Krebs MG, Hou J-M, Ward TH, Blackhall FH, Dive C. Circulating tumour cells: their utility in cancer management and predicting outcomes. Therapeutic Advances in Medical Oncology [Internet]. Therapeutic Advances in Medical Oncology (2010);2(6):351–65.

2Wu C-Y, Fu J-Y, Wu C-F, Hsieh M-J, Liu Y-H, Liu H-P, Hsieh JC-H, Peng Y-T. Malignancy Prediction Capacity and Possible Prediction Model of Circulating Tumor Cells for Suspicious Pulmonary Lesions. Journal of Personalized Medicine. (2021); 11(6):444.

3Lee, CH., Hsieh, J.CH., Wu, T.MH. et al. “Baseline circulating stem-like cells predict survival in patients with metastatic breast Cancer”. BMC Cancer (2019); 19:1167 

4Yoon, Hyeun Joong, et al. "Sensitive capture of circulating tumour cells by functionalized graphene oxide nanosheets." Nature nanotechnology 8.10 (2013): 735-741.

5Wu, Xiaoxia, et al. "Improved SERS nanoparticles for direct detection of circulating tumor cells in the blood." ACS applied materials & interfaces 7.18 (2015): 9965-9971.

6Wang, Shan-Shan, et al. "Direct plasmon-enhanced electrochemistry for enabling ultrasensitive and label-free detection of circulating tumor cells in blood." Analytical chemistry 91.7 (2019): 4413-4420.

  1. I believe that the CTCs value is important, but I suspect that the rate of change in the CTCs value before and after nCRT may be more important as a predictor of the effect of nCRT. Thank you in advance.

Thank you for this comment. We totally agree with you that the dynamic change in CTCs will be of great value, but unfortunately, CTCs at diagnosis are not available for this study group but we hope to include more samples in future analyses. We have added this to our limitation section.

  1. Table 1: What does y stand for in ypT and ypN? Please let us know.

When a patient has received preoperative therapy, the pathologic categories include a prefix “y” before TNM, thus “ypT” and “ypN” for pathologic classifications.

Reviewer 3 Report

The article focuses on the clinical utility of CTC as a biomarker in locally advanced esophageal cancer patients who have undergone surgery after preoperative treatment. This article might need modification or addition related to the rationale of MPR(MaHR) as an outcome of this study.  

Major Comments

1. I agree that pCR cases are candidates who avoid surgery. Still, I can't entirely agree that MPR(MaHR) cases are candidates who avoid surgery due to the pathological presence of cancer. Therefore, I think if the authors insist on the "surgery as needed" approach in this article,  the authors should evaluate the prediction of pCR by using cCR and CTCs. 

Author Response

AUTHORS’ RESPONSES TO COMMENTS FROM REVIEWER #3
The article focuses on the clinical utility of CTC as a biomarker in locally advanced esophageal cancer patients who have undergone surgery after preoperative treatment. This article might need modification or addition related to the rationale of MPR(MaHR) as an outcome of this study.  

Major Comments

I agree that pCR cases are candidates who avoid surgery. Still, I can't entirely agree that MPR(MaHR) cases are candidates who avoid surgery due to the pathological presence of cancer. Therefore, I think if the authors insist on the "surgery as needed" approach in this article, the authors should evaluate the prediction of pCR by using cCR and CTCs. 

Thank you for this comment. We agree that patients with any pathological presence of cancer must eventually be operated according to the current guidelines and the ultimate goal is to select patients with a pCR. However, our current CTC quantification/searching method is not advanced enough yet to detect cancer free patients (pCR). For esophageal cancer, the prediction of pCR after nCRT is an unfulfilled demand but serological biomarkers are starting to show promising results. When the accuracy of clinical/serological tools becomes precise enough to detect the clinical difference between MaHR and pCR, we agree that the focus should shift to patients with a pCR.

In the meantime, the SANO trial is investigating an active surveillance strategy for patients with a clinically complete response. If successful, patients with a “pCR” will be spared from surgery and patients with a MaHR will be safely detected during surveillance at a time when surgery still represents a curative option.

Round 2

Reviewer 3 Report

I have concerns about the tendency of this study toward overinterpreting data without important analysis. 

Author Response

We appreciate your comment. From a surgeon’s perspective, "surgery as needed" proposals lack promising biomarkers to aid the decision, which can be especially difficult when patients with good responses cannot be identified just by clinical assessment tools. In this study, liquid biopsy seems to provide objective evidence to help in the prediction of tumor response after neoadjuvant concurrent chemoradiotherapy. We hope this rationale and advances can be published, even though small, for other scientists to validate and inspire larger-scale clinical studies.